# A Comprehensive Analysis of Plasma Cytokines and Metabolites Shows an Association between Galectin-9 and Changes in Peripheral Lymphocyte Subset Percentages Following Coix Seed Consumption

**DOI:** 10.3390/nu14091696

**Published:** 2022-04-19

**Authors:** Yoshio Suzuki, Taisei Miyahara, Minami Jinnouchi, Yoshiki Miura, Hikari Taka, Naoko Kaga, Akiko Ohara-Takada

**Affiliations:** 1Graduate School of Health and Sports Science, Juntendo University, Inzai 270-1695, Japan; tl_bpnpne_22@yahoo.co.jp (T.M.); j.minami.67@gmail.com (M.J.); 2Laboratory of Proteomics and Biomolecular Science, Biomedical Research Core Facilities, Juntendo University Graduate School of Medicine, Tokyo 113-8421, Japan; y-miura@juntendo.ac.jp (Y.M.); takah@juntendo.ac.jp (H.T.); kagan@juntendo.ac.jp (N.K.); 3Research Center of Genetic Resources, National Agriculture and Food Research Organization, Tsukuba 305-8602, Japan; aohara@naro.affrc.go.jp

**Keywords:** metabolome, adlay, pearl barley, gut microbiota, omics

## Abstract

We previously reported that healthy adult males who consumed coix seeds for 1 week demonstrated an increased intestinal abundance of *Faecalibacterium prausnitzii* and altered peripheral lymphocyte subset percentages. However, the mechanism underlining these effects has not been elucidated. Therefore, cytokines and metabolites in plasma obtained in this study are comprehensively analyzed. A total of 56 cytokines and 52 metabolites in the plasma are quantified. Among them, 14 cytokines and 9 metabolites show significant changes in their levels following coix seed consumption. We examine the relationship between these changes and those in peripheral lymphocyte subset percentages and intestinal abundance of *F. prausnitzii*, which is also considerably altered following coix seed consumption. The galectin-9 concentration considerably decreased after coix seed consumption, and these changes correlate with those in cytotoxic T cells and pan T cells. Therefore, galectin-9 is possibly involved in the changes in peripheral lymphocyte subset percentages induced by coix seed consumption.

## 1. Introduction

Coix (*Coix lacryma-jobi* Linné. var. *mayuen* Stapf), also called Job’s tears or adley, is a tall, grain-bearing perennial plant belonging to the grass family *Poaceae*. In Japan, the polished endosperm of coix is approved for use as an ethical drug, referred to as “Yokuinin”, which is used for the treatment of verruca vulgaris. However, unexpectedly, only a few randomized clinical trials have clearly demonstrated its efficacy. Moreover, its mechanism and active ingredients have not been elucidated. A comprehensive review of the existing literature [1] has revealed that a few studies have demonstrated the promotion of spontaneous wart regression by this drug in human papillomavirus infection [2]. Moreover, injections of purified oils from coix seed have been shown to enhance the effects of cancer therapy in China [3,4,5,6]. In addition, coix-derived compounds, including coixenolide [7,8] and coixol [9,10,11], have been reported. Coixenolide has been reported to suppress the decrease in regulatory T cell levels in a mouse arthritis model [12]. Coixol is reported to have various activities, e.g., anti-tumor activity in vitro [13,14,15], reproductive trigger in rodents [16], enhancing insulin secretion in vitro [17,18], and enhancing airway mucin production and secretion in vitro [19]. Recently, coixol has been reported to modulate the immune response in lipopolysaccharide (LPS)-stimulated RAW264.7 cells [11]. However, the effects of both coixenolide and coixol on the human immune system have not been examined [1].

In a previous study [20], we investigated the effect of coix seed consumption on the immune systems of healthy individuals. When healthy adult males consumed 160 g/day of cooked coix seed, the abundance of *Faecalibacterium prausnitzii* increased in the gut microbiota. Moreover, the circulating killer T cell (CD3^+^ and CD8^+^ T cells), helper T cell (CD4^+^ T cells), and regulatory T cell (CD4^+^ and CD25^+^ T cells) percentages increased, whereas the natural killer (NK) cell (CD3^−^ and CD56^+^ cells) and memory T cell (CD3^+^, CD45RA^−^, CD45RO^+^) percentages decreased in the peripheral blood [20]. A meta-analysis showed a decrease in the intestinal *F. prausnitzii* in patients with inflammatory bowel disease [21]. A decrease in *F. prausnitzii* was also reported in patients with Crohn’s disease [22], colitis [23], *Clostridium difficile* infection [24], human immunodeficiency virus infection [25], and hepatitis B infection [26]. However, the relationship between the intestinal abundance of *F. prausnitzii* and peripheral lymphocyte subset percentages remains to be elucidated.

We hypothesized that the stimulation of the intestinal tract by coix seed increases the abundance of *F. prausnitzii* and affects the peripheral lymphocyte subset percentages by humoral factors via blood circulation. To clarify this hypothesis, plasma metabolites and cytokines collected before and after coix seed consumption in the previous study were comprehensively analyzed.

## 2. Materials and Methods

### 2.1. Plasma Samples

The plasma samples collected in the previous study [20] were analyzed. Briefly, a total of 18 healthy adult males were randomly allocated to either the coix seed consumption group (CS; *n* = 10) or the control group (CN; *n* = 8). The participants in the CS group consumed 160 g of cooked coix seed [27] daily for 7 days. Their plasma samples were collected before (pre) and after (post) coix seed consumption and stored at −80 °C until analysis.

The participants’ diet was Japanese style; participants in CS group consumed cooked coix seed substitute for cooked rice as staple food. The mean (SD) protein:fat:carbohydrate (P:F:C) ratio was 16.4% (3.3%): 29.3% (7.3%): 54.4 (10.5%) for the CS group while that for the CN group was 13.9% (3.1%): 27.4% (14.1%): 58.7% (16.8%), with no group differences observed [20]. In addition, as one of the exclusion criteria was receiving treatment or prescribed medication from a physician [20], no participants took any prescribed drugs, including antibiotics. However, one participant in CS group reported habitually taking the commercial over-the-counter probiotic agent “Shin-Biofermin S” (Taisho Pharmaceutical, Tokyo, Japan), which contains *Enterococcus faecalis*, *Lactobacillus acidophilus*, and *Bifidobacterium bifidum*. However, the participant’s microbiota data were excluded from the analysis because of missing data [20].

All participants provided written informed consent for participation. This study was approved by the Ethics Committee of Juntendo University Graduate School of Health and Sports Science (approval number: 31–72), and the study was performed in accordance with the ethical standards of the 1964 Declaration of Helsinki and its later amendments or comparable ethical standards. This study was registered in the UMIN Clinical Trials Registry (ID: UMIN000038831) before the initiation of interventions.

### 2.2. Cytokine Analysis

In our previous report [20], a total of 12 plasma cytokines were analyzed using the LEGENDplex^™^ Human Th Cytokine Panel (BioLegend, San Diego, CA, USA). This panel was used to quantify 12 human cytokines, including interleukin-2 (IL-2), IL-4, IL-5, IL-6, IL-9, IL-10, IL-13, IL-17A, IL-17F, IL-22, interferon-γ, and tumor necrosis factor-α, which are collectively secreted by T helper (Th)1, Th2, Th9, Th17, and Th22 cells.

Additional 44 plasma cytokines were analyzed using the LEGENDplex^™^ Th Cytokine Panels (BioLegend), including the Human Cytokine Panel 2 (13 cytokines), Human CD8/NK Panel (6 cytokines), Checkpoint Panel 1 (12 cytokines), and Human Proinflammatory Chemokine Panel 1 (13 cytokines). The cytokines were analyzed using each kit and the characteristics of each kit are summarized in Appendix A. All analyses were conducted at a private clinical laboratory (IMUH, Tokyo, Japan). In cases where the biomarker was undetected, one-half of the detection limit of the respective cytokine was used as the concentration.

### 2.3. Metabolite Analysis

Plasma metabolites were extracted and analyzed using the method recommended by Human Metabolome Technologies, Inc. (Tsuruoka, Japan). Briefly, methanol containing Internal Standard Solution 1 (H3304–1002, Human Metabolome Technologies, Inc.) was added to the plasma samples and mixed well. Subsequently, Milli-Q water and chloroform were added and centrifuged at 2300× *g* for 5 min at 4 °C. The upper aqueous layer was collected and filtered using a Millipore 5-kDa cutoff filter and dried. The metabolites were resuspended in 25 µL Milli-Q water containing Internal Standard Solution 3 (H3304–1004, Human Metabolome Technologies, Inc.) and used for capillary electrophoresis–mass spectrometry (CE–MS) analysis. All CE–MS experiments were performed using the Agilent 7100 CE capillary electrophoresis system (Agilent Technologies, Waldbronn, Germany) connected to the Agilent 6530 Accurate Quadrupole Time-of-Flight MS system (Agilent Technologies, Palo Alto, CA, USA). Each metabolite was identified and quantified based on the peak information, including mass-to-charge ratio, migration time, and peak area, using Profinder software, version B.08.00 (Agilent Technologies, Palo Alto, CA, USA). If a metabolite was undetected, one-half of the minimum concentration detected by the respective metabolite was used as the concentration.

### 2.4. Statistical Analysis

The Wilcoxon signed-rank test was used for the pre-intervention versus post-intervention comparisons of cytokines and metabolites. The standardized effect sizes for the CS group (*n* = 10) and the CN group (*n* = 8) to satisfy α = 0.05 (two tails) and power (1−β) = 0.8 were respectively 1.03 and 1.19 according to G*Power ver.3.1.9.2 [28,29]. MetaboAnalyst 5.0 (https://www.metaboanalyst.ca/MetaboAnalyst/home.xhtml, accessed on 20 February 2022) was used for statistical and pathway analyses. The relationships between two parameters were determined using Spearman’s correlation coefficients, which were calculated using SPSS, version 23 (IBM Corporation, Tokyo, Japan). A *p*-value of < 0.05 was considered statistically significant.

## 3. Results

### 3.1. Cytokines

Altogether, 14 cytokines in the CS group and 5 cytokines in the CN group showed significant changes following coix seed consumption (Table 1, Appendix A).

We examined the relationships between the changes in the cytokine concentrations ([post-intervention—pre-intervention]/pre-intervention) and the changes in the peripheral lymphocyte subset percentages (post-intervention—pre-intervention), which also changed significantly after coix seed consumption.

Galectin-9 showed a significant positive correlation with the NK cell percentage (R = 0.616, *p* = 0.006) and a negative correlation with the killer T cell (R = −0.439, *p* = 0.069), Th cell (R = −0.375, *p* = 0.126), and regulatory T (Treg) cell (R = −0.181, *p* = 0.473) percentages. A significant negative correlation was also observed between galectin-9 and the pan T cell (CD3^+^ cell) percentage (R = −0.509, *p* = 0.031) (Figure 1).

The cytotoxic T cell percentage showed a negative correlation with PD ligand 2 (PD-L2) (R = −0.538, *p* = 0.021) and the soluble IL-2 receptor (sCD25) (R = −0.567, *p* = 0.014). The NK cell percentage showed a positive correlation with lymphocyte-activation gene 3 (LAG-3) (R = 0.554, *p* = 0.017). Significant positive correlations were also observed between galectin-9 and sCD25 (R = 0.633, *p* = 0.005) and between PD-L2 and LAG-3 (R = 0.484, *p* = 0.042) (Figure 2).

The memory T cell percentage showed a positive correlation with macrophage inflammatory protein (MIP)-1α (R = 0.539, *p* = 0.021); however, the correlation between the memory T cell percentage and MIP-1β was not significant (R = −0.100, *p* = 0.693). MIP-1α showed a significant positive correlation with MIP-1β (R = 0.504, *p* = 0.034) (Figure 3).

The change in the abundance of *F. prausnitzii* (post-intervention—pre-intervention), which was significantly increased after coix seed consumption, showed a significant positive correlation with growth-regulated oncogene (GRO)-α, which was significantly decreased after coix seed consumption (R = 0.525, *p* = 0.031). In addition, GRO-α showed a negative, but nonsignificant, correlation with the NK cell percentage (R = −0.465, *p* = 0.058). This relationship was significant in the CN group (R = −0.833, *p* = 0.010) but not in the CS group (R = 0.127, *p* = 0.726) (Figure 4).

### 3.2. Metabolites

Fifty-two metabolites were detected and quantified in total. Among them, compared with the pre-intervention, nine metabolites in the CS group and seven metabolites in the CN group showed significant changes post-intervention (Table 2 and Appendix A).

We examined the relationships between the changes in metabolites ([post-intervention—pre-intervention]/pre-intervention) and the changes in the lymphocyte subset percentages (post-intervention—pre-intervention), which were significantly altered by coix seed consumption. A significant negative correlation was only observed between the NK cell percentage and isocitric acid (R = −0.474, *p* = 0.047) (Figure 5). Similarly, the relationships between the changes in metabolites and the change in the abundance of *F. prausnitzii* (post-intervention—pre-intervention) were examined, but no significant correlation was observed.

In terms of the correlations between cytokines and metabolites, which showed significant changes with coix seed consumption, a significant negative correlation was only observed between isocitric acid and galectin-9 (R = −0.523, *p* = 0.026).

A pathway analysis of the nine plasma metabolites showed significant changes after coix seed consumption was conducted against the homo sapiens library (Kyoto Encyclopedia of Genes and Genomes). Three metabolites, including isocitric acid, cis-aconitic acid, and glyoxylic acid, were mapped to glyoxylate and dicarboxylate metabolism. No other pathway was found to be mapped to more than two metabolites.

## 4. Discussion

Galectin-9 is one of the β-galactoside-binding mammalian lectins found in the mouse embryonic kidney [30]. It causes dose-dependent apoptosis of mouse thymocytes [31] and also induces apoptosis in lymphocyte cell lines, including MOLT-4 (T cells), Jurkat (T cells), BALL-1 (B cells), THP-1 (monocytes), and HL-60 (myelocytes) cells [32]. A previous study using CD4^+^ and CD8^+^ T cells isolated from human peripheral blood showed that apoptosis is more strongly induced in activated cells than in non-activated cells, as well as in CD4^+^ cells than in CD8^+^ cells [32]. In addition, galectin-9 induces Th1 cell apoptosis but not Th2 cell apoptosis in mice [33]. Recently, Yang et al., reported that anti-galectin-9 therapy increased the number of cytotoxic CD8^+^ T cells and immunosuppressive Treg cells [34]. As described above, galectin-9 suppresses Th cells and killer T cells. In this study, the galectin-9 concentration significantly decreased with coix seed consumption, and this change showed a negative correlation with the cytotoxic T cell percentage and a positive correlation with the NK cell percentage. Although not significant, a negative correlation was observed between the changes in the Th cell and Treg cell percentages. Therefore, coix seed was considered to increase the cytotoxic T cell, Th cell, and Treg cell percentages by decreasing the plasma galectin-9 concentration and indirectly decreasing the ratio of NK cells.

sCD25, which is otherwise known as the soluble IL-2 receptor (IL-2R), is generated by the cleavage of the membrane-bound IL-2Ra [35]. sCD25 may act as an antagonist of the IL-2 by neutralizing it [36], whereas sCD25 promotes the proliferation of T cells purified from peripheral blood mononuclear cells [37].

The effect of sCD25 on the phosphorylation of the signal transducer and activator of transcription 5 has been reported to be inhibited [38] and promoted [39] by IL-2. Thus, the physiological role of sCD25 has not been completely clarified [40].

PD-L2, which is a cell surface protein belonging to the B7-CD28 family [41], is one of the two ligands that bind to the immune checkpoint receptor PD-1 [42]. The serum concentrations of PD-L2 are proposed to be biomarkers for esophageal cancer [43] and pancreatic cancer [44], but their role in circulation has not been clarified.

LAG-3 is a transmembrane protein that belongs to the immunoglobulin superfamily [45]. It is involved in the negative regulation of cell proliferation and activation [46]. The serum concentration of LAG-3 is a potential prognostic marker for gastric cancer [47], non-small-cell lung cancer [48], and hepatocellular carcinoma [49]. However, its physiological role is still unknown.

As described above, sCD25, PD-L2, and LAG-3 are all membrane-bound proteins; however, their physiological importance in circulation is not clear. In this study, we found that the plasma concentration of these cytokines was considerably altered by coix seed consumption, and their rate of change in concentration showed a significant correlation with the change in cytotoxic T cell and NK cell percentages, which were also significantly altered by coix seed consumption. In addition, sCD25 showed a positive correlation with galectin-9, and PD-L2 showed a positive correlation with LAG-3. Therefore, sCD25, PD-L2, and LAG3, either alone or in combination with galectin-9, may affect the percentage of circulating cytotoxic T cells, NK cells, Th cells, and Treg cells.

GRO-α, which is also known as CXC chemokine ligand 1, is a chemokine that binds to the IL-8 receptor [50]. In this study, GRO-α was significantly decreased by coix seed consumption, and this change showed a positive correlation with the change in the abundance of *F. prausnitzii*, which was significantly increased by coix seed consumption. However, the direction of this correlation was opposite to the direction of change; thus, this correlation may have been observed only by chance.

In this study, 9 of the 52 detected metabolites showed significant changes upon coix seed consumption. However, we could not identify a clear relationship between these changes and the changes in the peripheral lymphocyte subset percentages and the intestinal abundance of *F. prausnitzii*, which were also significantly altered by coix seed consumption. Therefore, the changes in the lymphocyte subset percentages do not seem to be driven by metabolic changes.

This study has some limitations. First, the correlations were discussed, considering the change in peripheral lymphocyte subset percentage is driven by the change in intestinal abundance of *F. prausnitzii* via circulation. However, the correlation does not indicate the direction of change or the cause–effect relationship. Therefore, further studies are warranted to elucidate the mechanisms underlying the associations observed in this study. Second, the sample size of this study was small, as reported previously [20]. The results of this research should be confirmed by further studies.

## 5. Conclusions

In this study, we conducted a comprehensive analysis of cytokines and metabolites in the plasma samples obtained before and 1 week after coix seed consumption and examined their relationships with changes in the peripheral lymphocyte subset percentages and the intestinal abundance of *F. prausnitzii*, which also showed significant changes with coix seed consumption. Among the 14 cytokines and 9 metabolites that demonstrated a significant change with coix seed consumption, galectin-9 concentration significantly decreased, and this change was correlated with the changes in cytotoxic T cell and pan T cell percentages. Therefore, galectin-9 may be involved in the changes in the peripheral lymphocyte subset percentages induced by coix seed consumption.

## Figures and Tables

**Figure 1 nutrients-14-01696-f001:**
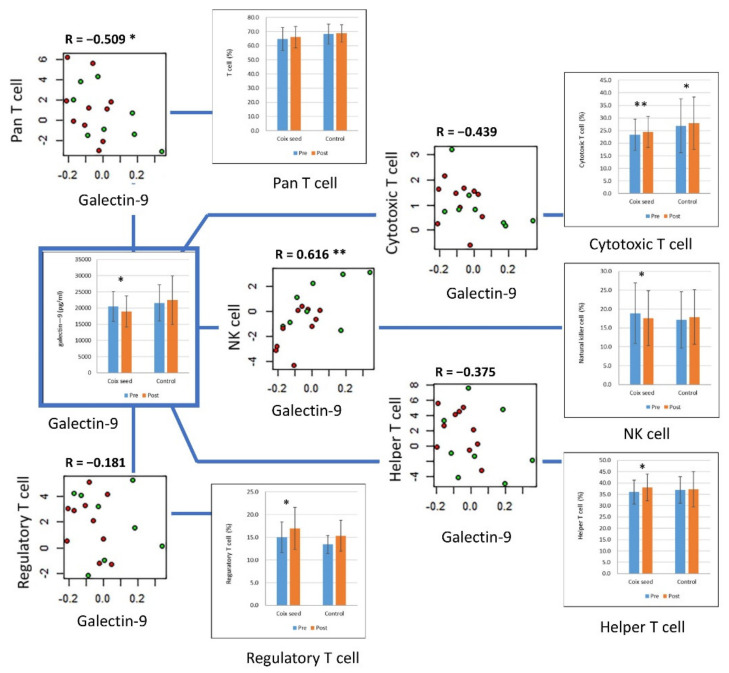
Galectin-9 and its correlation with peripheral lymphocyte subset percentages. The plasma concentration of galectin-9 significantly decreased after coix seed consumption. The change in galectin-9 showed a significant correlation with the changes in natural killer (NK) cell and pan T cell (CD3^+^ cell) percentages. The correlations between galectin-9 and cytotoxic T cells, helper T cells, and regulatory T cells are also presented. In the scatter plots, the red and green circles denote the coix seed consumption group and the control group, respectively. In the bar graphs, the blue and orange bars denote before (pre-intervention) and after (post-intervention) coix seed consumption, respectively. *, *p* < 0.05; **, *p* < 0.01.

**Figure 2 nutrients-14-01696-f002:**
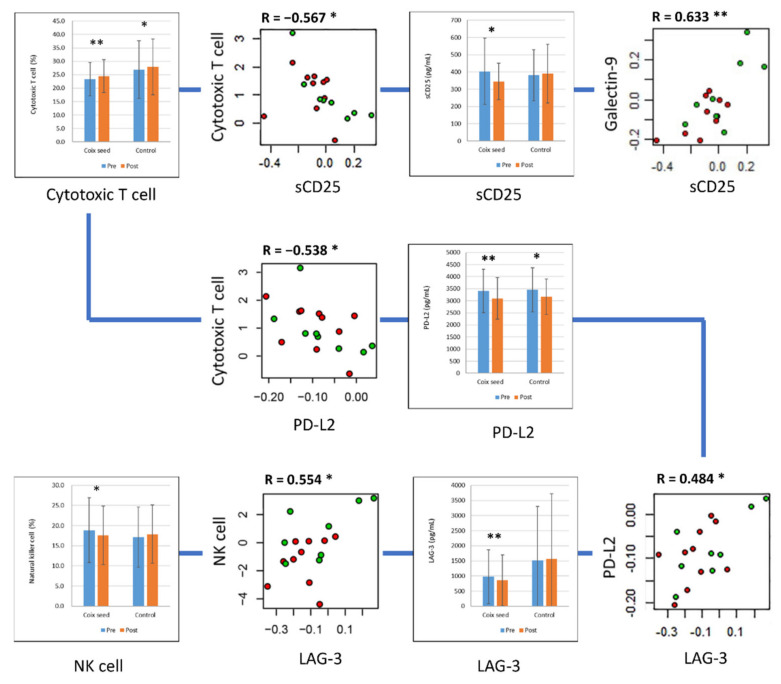
sCD25, PD-L2, and LAG-3 and their correlation with peripheral lymphocyte subset percentages. The plasma concentration of sCD25, PD-L2, and LAG-3 significantly decreased after coix seed consumption. The changes in sCD25 and PD-L2 showed a significant correlation with the changes in cytotoxic T cell percentage; sCD25 also showed significant correlation with plasma concentration of galectin-9. The change in LAG-3 showed a significant correlation with the changes in natural killer (NK) cell percentage and plasma concentration of PD-L2. In the scatter plots, the red and green circles denote the coix seed consumption group and the control group, respectively. In the bar graphs, the blue and orange bars denote before (pre-intervention) and after (post-intervention) coix seed consumption, respectively. *, *p* < 0.05; **, *p* < 0.01.

**Figure 3 nutrients-14-01696-f003:**
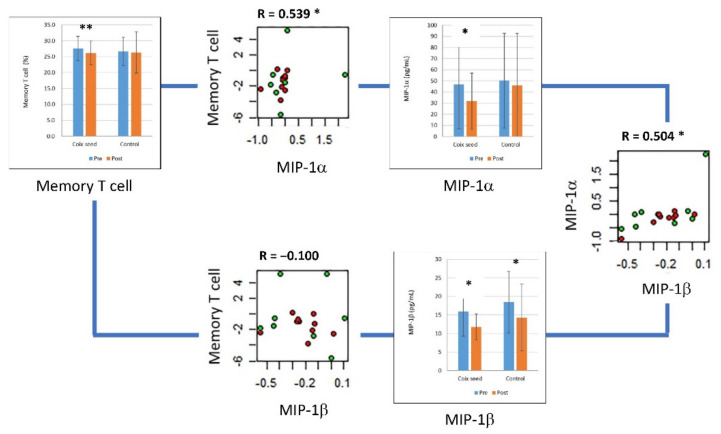
MIP-1α, MIP-1β, and their correlation with the peripheral memory T cell percentage. The plasma concentration of MIP-1α and MIP-1β significantly decreased after coix seed consumption. The change in MIP-1α showed a significant correlation with the change in memory T cell percentage. The changes in MIP-1α and MIP-1β were significantly correlated. In the scatter plots, the red and green circles denote the coix seed consumption group and the control group, respectively. In the bar graphs, the blue and orange bars denote before (pre-intervention) and after (post-intervention) coix seed consumption, respectively. *, *p* < 0.05; **, *p* < 0.01.

**Figure 4 nutrients-14-01696-f004:**
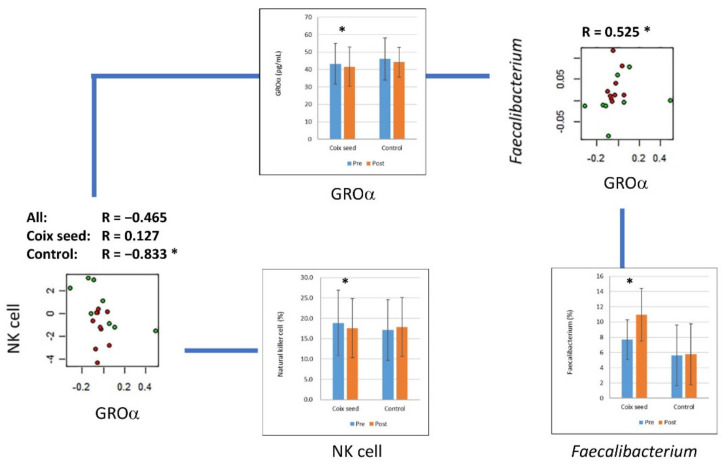
GRO-α and its correlation with the natural killer cell percentage and intestinal abundance of *Faecalibacterium prausnitzii*. The plasma concentration of GRO-α significantly decreased after coix seed consumption. The change in GRO-α showed a significant correlation with the change in intestinal abundance of *F. prausnitzii*. The change in GRO-α showed a negative correlation with the change in natural killer (NK) cell percentage; however, the correlation differed between the coix seed consumption group and the control group. In the scatter plots, the red and green circles denote the coix seed consumption group and the control group, respectively. In the bar graphs, the blue and orange bars denote before (pre-intervention) and after (post-intervention) coix seed consumption, respectively. *, *p* < 0.05.

**Figure 5 nutrients-14-01696-f005:**
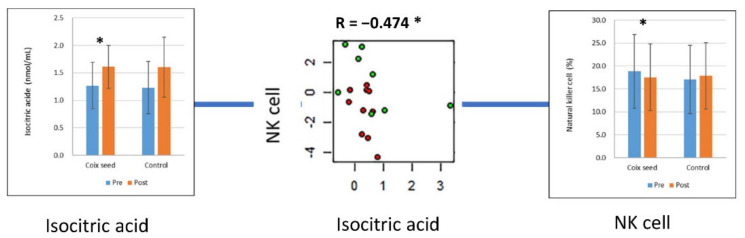
Isocitric acid and its correlation with the natural killer cell percentage. The plasma concentration of isocitric acid significantly increased with coix seed consumption. The change in isocitric acid showed a significant correlation with the change in natural killer (NK) cell percentage. In the scatter plot, the red and green circles denote the coix seed consumption group and the control group, respectively. In the bar graphs, the blue and orange bars denote before (pre-intervention) and after (post-intervention) coix seed consumption, respectively. *, *p* < 0.05.

**Table 1 nutrients-14-01696-t001:** Plasma cytokines that showed significant changes after coix seed consumption.

Cytokine	Group	Pre		Post		*p*	
		Mean	SD	Mean	SD		
TNF-α	Coix seed	6.792	3.568	2.704	2.673	0.0020	**
	Control	9.527	5.652	2.69	2.366	0.0234	*
PD-L2	Coix seed	3409.385	897.79	3100.019	864.241	0.0024	**
	Control	3452.927	910.039	3168.057	727.724	0.0422	*
LAG-3	Coix seed	974.04	887.598	858.48	843.609	0.0059	**
	Control	1523.65	1782.27	1567.275	2148.765	0.7422	
Eotaxin	Coix seed	26.856	5.633	25.317	5.982	0.0063	**
	Control	27.711	9.639	25.637	9.06	0.1025	
sCD27	Coix seed	16,901.79	5130.348	15,185.19	5108.296	0.0087	**
	Control	16,075.838	4472.859	14,340.6	2940.917	0.0645	
IL-8	Coix seed	45.01	42.048	23.249	12.161	0.0098	**
	Control	51.822	41.884	37.705	49.843	0.0391	*
Granulysin	Coix seed	1362.035	584.072	1236.16	566.357	0.0177	*
	Control	1251.207	353.024	1195.626	278.356	0.3960	
MIP-1β	Coix seed	15.962	6.686	11.808	3.477	0.0208	*
	Control	18.446	8.274	14.329	8.97	0.0341	*
GROα	Coix seed	43.306	11.6	41.594	11.222	0.0268	*
	Control	46.075	12.12	44.224	8.499	0.4609	
sCD25	Coix seed	403.288	192.1	344.777	106.083	0.0371	*
	Control	381.585	148.286	390.705	169.936	0.7309	
Galectin-9	Coix seed	20,522.77	4604.374	18,982.87	4809.372	0.0390	*
	Control	21,600.662	5630.121	22,487.975	7507.335	0.5870	
MIP-1α	Coix seed	46.864	39.994	31.732	25.262	0.0422	*
	Control	50.209	42.623	45.774	46.875	0.6726	
RANTES	Coix seed	1589.879	534.273	1393.553	465.832	0.0488	*
	Control	2475.471	1999.592	1363.781	612.868	0.0656	
B7.2	Coix seed	119.586	31.025	112.681	29.667	0.0497	*
	Control	104.954	52.349	104.704	64.895	0.4609	

Concentrations are measured in pg/mL. *, *p* < 0.05; **, *p* < 0.01.

**Table 2 nutrients-14-01696-t002:** Plasma metabolites that showed significant changes after coix seed consumption.

Metabolite	Group	Pre		Post		*p*	
		Mean	SD	Mean	SD		
Malic acid	Coix seed	3.227	0.679	4.204	0.76	<0.0001	****
	Control	3.183	0.403	4.617	1.337	0.0369	*
Creatinine	Coix seed	66.355	8.19	59.674	5.94	0.0017	**
	Control	64.205	8.077	60.045	10.131	0.0414	*
cis-Aconitic acid	Coix seed	1.041	0.344	1.411	0.339	0.0023	**
	Control	1.018	0.386	1.415	0.462	0.1386	
Anthranilic acid	Coix seed	0.323	0.164	0.148	0	0.0084	**
	Control	0.35	0.358	0.148	0	0.1558	
Cysteine	Coix seed	3.932	1.159	2.625	1.049	0.0144	*
	Control	3.63	0.697	2.19	1.345	0.0582	
4-Aminobutyric	Coix seed	9.284	6.928	11.157	7.594	0.0239	*
acid (GABA)	Control	10.46	5.78	12.469	6.57	0.1443	
Citrulline	Coix seed	26.376	6.821	29.838	5.495	0.0241	*
	Control	28.127	4.522	28.002	5.617	0.9536	
Isocitric acid	Coix seed	1.27	0.425	1.611	0.393	0.0251	*
	Control	1.232	0.476	1.602	0.547	0.2261	
Glyoxylic acid	Coix seed	40.741	17.426	53.421	16.734	0.0490	*
	Control	34.722	9.372	53.225	15.079	0.0391	*

Concentrations are measured in nmol/mL. *, *p* < 0.05; **, *p* < 0.01, ****, *p* < 0.0001.

## Data Availability

The data are not publicly available for privacy and ethical reasons. Those who wish to obtain the data published in this study may contact the corresponding author.

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
