# Peer review of "A Comprehensive Analysis of Plasma Cytokines and Metabolites Shows an Association between Galectin-9 and Changes in Peripheral Lymphocyte Subset Percentages Following Coix Seed Consumption"

_nutrients, 2022, doi:10.3390/nu14091696_

Round 1

Reviewer 1 Report

The manuscript "Comprehensive analysis of plasma cytokines and metabolites shows an association between galectin-9 and changes in peripheral lymphocyte subset percentages following coix seed consumption" by Yoshio Suzuki et al. is indeed a comprehensive analysis of 56 cytokines and 52 metabolites in the plasma samples collected previously from 18 healthy adult males, 10 of which consumed cooked coix seed for 7 days. The authors investigated the hypothesis that stimulation of the intestinal tract by coix seed increases the abundance of F. prausnitzii and affects peripheral lymphocyte subset percentages by humoral factors within blood circulation. This study revealed 14 cytokines and 9 metabolites that demonstrated a significant change with coix seed consumption,, however,  galectin-9 was the sole one that displayed significantly decreased concentration that was correlated with changes in cytotoxic T cell and pan T cell percentages. No other clear relationship between changes in peripheral lymphocyte subset percentages and the intestinal abundance of F. prausnitzii was observed. 

The manuscript is very well-written and the results are presented in a clear and concise way. Overall the quality of presentation is high and the conclusions are supported by the experimental findings. Importantly, the limitations of this study are clearly documented in the last paragraph. I would be really satisfied if this study included a more significant sample size, however, I reckon that the findings of this study will stimulate further research in the field.

Although the overall merit of this work would be higher if the authors had studied more samples, the high quality of the experimental work, analysis of and presentation of the results renders this manuscript of merit for publication to Nutrients.

My only suggestion is that the authors expand the first paragraph to include more information about coix-derived compounds and their potential therapeutic applications.

Author Response

I appreciate your kind instruction. Our response is as attached.

Reviewer 2 Report

In the present paper, Yoshio Suzuki and colleagues investigated if the stimulation of the intestinal tract by coix seed increases the abundance of F. prausnitzii and affects peripheral lymphocyte subset percentages by humoral factors via blood circulation. To clarify this hypothesis, plasma metabolites and cytokines collected before and after coix seed consumption in a previous study were comprehensively analyzed. The authors found that Galectin-9 concentration considerably decreased after coix seed consumption, and these changes were correlated with those in cytotoxic T cells and pan T cells. Therefore, they concluded that galectin-9 is possibly involved in the changes in peripheral lymphocyte subset percentages induced by coix seed consumption. Overall, I think that the paper could be of interest for readers of Nutrients and researchers, in general, on a current topic of research.

I make some suggestions for improve the quality of the manuscript.

1) Please better define and discuss the power analysis of study.

2) Have you analyzed the nutritional status of patients included in this analysis? Please explain. The authors, if possible, should incorporate in tables the dietary pattern of the patients included in the present study (i.e. Mediterranean-style diet, Plants-based diet, Nordic dietary pattern, etc.) and analyze their findings considering this crucial aspect. In this way, I feel that the readers can better understand the intriguing results obtained in the present clinical study and their possible application to clinical practice.

3) Some medicines/drugs could be used by patients (Antibiotics? Probiotics?). This aspect could interfere with results here revealed. Please better clarify this aspect and eventually take this factor into account in multiple logistic regression analysis.

Author Response

(The authors gave the same response as above.)

Round 2

Reviewer 2 Report

Thank you for addressing my comments well. I have no further remarks.